# Accuracy of Imaging Scoring Indexes in Pediatric Crohn’s Disease Patients

**DOI:** 10.3390/biomedicines13051157

**Published:** 2025-05-09

**Authors:** Goran Hauser, Goran Palčevski, Barbara Čandrlić, Pero Hrabač, Damir Miletić

**Affiliations:** 1Department of Gastroenterology, Centre for Digestive Disorders, Clinical Hospital Centre Rijeka, Faculty of Medicine, University of Rijeka, 51000 Rijeka, Croatia; goran.hauser@medri.uniri.hr; 2Department of Pediatric Gastroenterology, Clinical Hospital Centre Rijeka, Faculty of Medicine, University of Rijeka, 51000 Rijeka, Croatia; goran.palcevski@uniri.hr; 3Department of Radiology, Clinical Hospital Centre Rijeka, Faculty of Medicine, University of Rijeka, 51000 Rijeka, Croatia; barbaracandrlic@yahoo.com (B.Č.); damir.miletic@medri.uniri.hr (D.M.); 4Department of Medical Statistics, Epidemiology and Medical Informatics, “Andrija Štampar” School of Public Health, School of Medicine, University of Zagreb, 10000 Zagreb, Croatia

**Keywords:** Crohn’s disease, pediatrics, capsule endoscopy, magnetic resonance imaging, disease activity index

## Abstract

**Background:** Crohn’s disease (CD) is a chronic inflammatory condition that can affect the gastrointestinal tract and cause significant extraintestinal manifestations. Diagnosing and monitoring disease activity, especially in pediatric patients, remains a challenge due to the variable clinical presentations and limitations of traditional imaging methods. **Objective:** This study aimed to evaluate and compare the diagnostic accuracy and clinical utility of small bowel capsule endoscopy (SBCE) versus magnetic resonance enterography (MRE) for assessing disease activity and extent in pediatric Crohn’s disease using the Pediatric Crohn’s Disease Activity Index (PCDAI) and Simple Endoscopic Score for Crohn’s Disease (SES-CD) as reference standards. **Methods:** In this prospective study, 52 pediatric patients with newly diagnosed CD underwent upper and lower endoscopy, MRE, and SBCE. The SBCE images were analyzed using the Capsule Endoscopy Crohn’s Disease Activity Index (CECDAI), while the MRE images were scored using the Crohn’s Disease MRI Index (CDMI). Correlations of these findings with PCDAI and SES-CD were statistically analyzed. **Results:** CECDAI and CDMI demonstrated strong correlations with PCDAI (r = 0.517 and r = 0.525, respectively; *p* < 0.001). The correlations between CECDAI and SES-CD were less pronounced but significant. SBCE and MRE showed comparable efficacy in detecting small bowel lesions, with both methods offering valuable insights into the disease status. **Conclusions:** SBCE is a reliable, non-invasive tool for diagnosing and monitoring pediatric CD, comparable to MRE. While SBCE offers higher resolution for mucosal evaluation, it requires additional expertise for optimal interpretation. The adoption of SBCE alongside MRE could enhance diagnostic accuracy and early therapeutic interventions for pediatric CD.

## 1. Introduction

Crohn’s disease (CD) can affect the gastrointestinal (GI) tract either partially or as a whole, with a wide range of extraintestinal manifestations. It has a chronic nature characterized by transmural inflammation and segmental skip lesions. The incidence has increased in the industrialized world and now ranges from 3 to 20 cases per 100,000 [1,2]. Risk factors include genetics, environmental factors, gut microbiota, and changes in the gut microbiome [3,4]. The clinical presentation is variable and includes a wide spectrum of gastrointestinal and systemic symptoms.

Uncontrolled inflammation and poor responsiveness to therapy can lead to long-term complications such as stricture and fistulae formation. In the past, clinical indices alone were mainly used to measure the response to therapy but studies have shown poor correlation between clinical and endoscopic measures [5,6].

Patients often present with diarrhea and abdominal pain, weight loss, fatigue, and fever. Bloody stool may occur in severe cases, but this symptom is more often associated with ulcerative colitis. The most common complications of pediatric Crohn’s disease are growth stagnation, malnutrition, and pubertal delay, and about 20% to 30% of all patients with IBD begin showing symptoms before the age of 18 years [7].

The laboratory findings in patients with CD are non-specific; thus, imaging techniques and endoscopy procedures play an important role in the diagnosis and evaluation of Crohn’s disease. Upper GI endoscopy and ileocolonoscopy are considered the gold standard for evaluating mucosal inflammation; however, a large part of the small intestine cannot be examined using these standard endoscopy procedures [8]. Since CD only affects the small intestine in about 30–40% of cases, standard endoscopy is not always the best diagnostic option [9]. In this context, small bowel capsule endoscopy (SBCE) is a viable alternative to standard imaging techniques for the small bowel. One of the main indications for video capsule endoscopy is suspected Crohn’s disease, and several studies have proven that capsule endoscopy is more reliable than other imaging techniques in detecting inflammation of the small bowel [10,11]. Mucosal healing has become an important treatment endpoint in Crohn’s disease patients, and SBCE may increase the detection of various lesions in the small bowel [12,13,14].

Another imaging method for evaluating the small bowel that provides views of all the layers of the bowel wall is magnetic resonance enterography (MRE). MRE is characterized by high sensitivity and specificity for Crohn’s disease [15,16]. Furthermore, MRE is accurate in the detection of extra-luminal complications of CD and strictures [17].

The aim of the present study was to compare and possibly correlate the findings obtained from MRE and SBCE with the Pediatric Crohn’s Disease Activity Index (PCDAI) at the time of disease presentation.

## 2. Subjects and Methods

### 2.1. Study Design

Pediatric Crohn’s disease patients (<18 years old) seen in the Gastroenterology and Hepatology Department of the Pediatric Clinic at the Clinical Hospital Centre Rijeka, Croatia, were considered for inclusion into the study. We undertook a prospective double-blinded comparison study to compare the efficacy of SBCE to MRE in pediatric patients in the detection of Crohn’s disease in the small bowel.

### 2.2. Participants

Pediatric patients with newly diagnosed Crohn’s disease admitted to the Department of Pediatrics in the Clinical Hospital Centre Rijeka from December 2012 to September 2020 were enrolled. All patients aged 18 years or younger with a clinical and laboratory suspicion for Crohn’s disease initially underwent upper and lower endoscopy under deep sedation. PCDAI and Simple Endoscopic Score for Crohn’s Disease (SES-CD) were both assessed at the time of admission. We enrolled a convenience sample of 35 boys and 17 girls, with a mean age of 14 years. All patients with suspected small bowel involvement after the endoscopy investigation underwent MRE and SBCE within a two-month period. The exclusion criteria were a known allergy to the contrast material, a finding of gut obstruction on MRE, or an inability to swallow the capsule. All SBCE examinations were successfully performed and reached the cecum. There were no complications regarding the SBCE procedure. The patients who met the eligibility criteria underwent MRI and SBCE within a four-week interval. Since the patient selection was essentially random, we only included the basic demographic data (sex and age) about our subjects.

### 2.3. Scoring Systems

The Pediatric Crohn’s Disease Activity Index (PCDAI) is a validated tool used to assess disease activity in children and adolescents with Crohn’s disease. It evaluates their clinical symptoms, weight, linear growth, physical findings, laboratory results, and general functioning over a 7-day period, providing a numerical score from 0 to 100 to classify disease severity [18].

The Simple Endoscopic Score for Crohn’s Disease (SES-CD) is an endoscopic scoring system designed to assess the severity of Crohn’s disease by evaluating four key endoscopic variables: ulcer size, proportion of ulcerated surface, extent of the affected surface, and presence of stenosis across five ileocolonic segments. The scores range from 0 to 56, with higher scores indicating more severe disease [19,20].

### 2.4. SBCE Technique

SBCE was performed with the PillCam™ SB 3 (Medtronic, Yokneam, Israel). The fasting period began the day before the examination. Bowel cleansing was achieved with 2.0 L of a polyethylene glycol solution, using a split regimen (Moviprep, Norgine Limited, Mid Glamorgan, UK), taken 16 h before the beginning of the examination. Capsules were administered at 8.30 a.m. following an overnight fast commencing at 10 p.m. on the night before SBCE. Liquids were allowed until eight hours before the examination.

Patients were permitted to drink fluids 4 h and eat solids 6 h after capsule ingestion and were instructed to return to the hospital no later than 10 h after the ingestion. Data from the recorder were downloaded on site within the Department of Gastroenterology and analyzed by one clinical SBCE expert blinded to the patient’s clinical information (Figure 1). The images were interpreted using Rapid Reader software (version 8.0, Given Diagnostic Imaging Systems). The data were evaluated using the Capsule Endoscopy Crohn’s Disease Activity Index (CECDAI or Niv score) [21]. The Niv score defines the severity and extent of the mucosal inflammatory process and stenosis for two segments of the small bowel: the proximal and distal segments. The final score is obtained by summing the proximal and distal scores and ranges from a minimum of 0 to a maximum of 36. Endoscopic remission correlates with a score of less than 4, while cases of active disease result in scores above 4 points [22].

This method carries a notable risk of capsule retention (CR). This risk can be reduced by employing patency capsules and pre-procedural imaging; however, if retention occurs, it can typically be managed conservatively, endoscopically, or through minimally invasive surgery [23].

### 2.5. MRE Technique

The MRE findings were evaluated using Crohn’s disease MRI index (CDMI) scores. The MRE images were read by two experienced investigators who independently reviewed all the patients.

The examination was performed using 1.5 tesla MRI system with built-in spine array coli and two flexible surface coils (Magnetom Avanto, Siemens, Erlangen, Germany). We performed the following sequences: HASTE (ST = 100 mm), true FISP: +FS/−FS, HASTE: +FS/−FS; ST = 6 mm, true FISP (axial), DWI, after contrast administration (Gadolinium), VIBE, true FISP, T1 postcontrast. Lesions were clearly displayed, allowing for scores to be calculated (Figure 2).

The patients underwent standard bowel cleansing within 24 h prior to examination (see above), and approximately 60 min prior to examination, we strongly encouraged the patients to drink at least 1000 mL of the oral contrast agent containing 0.5% mannitol and 0.5% milled carob. We used the prone position for better abdominal compression, thus reducing number of coronal slices and examination time.

After rectal introduction of Foley’s catheter, warm tap water was administered and filling of the colon was monitored using HASTE sequences obtained in the coronal plane. An intravenous spasmolytic agent (butylscopolamine 20 mg) was administered at the beginning of the water enema administration. Prior to gadolinium contrast application, butylscopolamine (20 mg i.v.) was repeatedly administered. Thereafter, a paramagnetic contrast agent (Magnevist, Bayer-Schering, Germany) was intravenously injected. The postcontrast VIBE sequence was repeated using the same acquisition parameters after a time delay of 75 s and two minutes.

One of the issues we faced while designing the study and later during the experimental phase was the comparability of lesion locations for the two techniques. While precise lesion positions can enhance correlation analyses, minor discrepancies in lesion localization generally do not significantly alter the comparative value of these techniques, especially when evaluating the overall disease extent, activity, and complications. Therefore, both modalities have practical clinical value despite potential variations in lesion localization. While it certainly makes sense to question the comparability of the spatial distributions of lesions detected by different techniques, our experience indicates that, statistically speaking, the probability of involvement of a specific bowel segment detected by various methods is comparable.

### 2.6. Data Collection and Interpretation

All patients were recruited at the Pediatric Department in the Clinical Hospital Centre Rijeka. All patients underwent MRE and after patency approval, they were referred to the Gastroenterology Department to perform SBCE within two weeks. The SBCE images were interpreted by one capsule reader, and MRE images were interpreted by an experienced gastrointestinal radiologist. The capsule reader and the radiologist both did not know the patients’ clinical data results or previous imaging findings.

The study conformed to the ethical guidelines of the 1975 Declaration of Helsinki’s Good Clinical Practice guidelines and ethical approval was obtained from our institutional review board prior to the commencement of the study (approval date: 25 May 2011; reference No.: 853.1). All subjects who participated in the study provided written informed consent. The study was registered at ClinicalTrials.gov under identifier NCT02006498. All authors had access to the study data and reviewed and approved the final manuscript.

### 2.7. Statistical Analysis

The statistical analysis was performed using the Statistica software package (Cloud software group, Fort Lauderdale, FLA, USA (2024); version 14; http://tibco.com). The normal distribution of the metric variables was tested using Shapiro–Wilk’s test so that the appropriate parametric or non-parametric tests could be used in the analysis. Correlations between metric variables were analyzed using Spearman’s method, while for multiple regression, the forward stepwise method was used.

Prior to the start of the study, a sample size calculation was performed in PASS (PASS 15 Power Analysis and Sample Size Software (2017); NCSS, LLC, Kaysville, UT, USA; www.ncss.com/software/pass, accessed on 9 December 2024). With the power set to 0.80, alpha (statistical significance) set to 0.05, and the aim to detect a (conservatively set) difference of at least 0.4 between the null hypothesis correlation of 0 and the alternative hypothesis correlation of 0.4 using a two-sided hypothesis test, the required sample size was 46. By the end of the study period, we managed to include a total of 52 subjects.

## 3. Results

In the present study, we enrolled 52 subjects: 35 boys and 17 girls. The mean age of the participants at the time of the exam was 14.0 years (SD = 2.34 years; range: 8–18 years). The ages of the boys and girls were comparable (*p* = 0.804). The mean PCDAI value was 21.3 (SD = 13.39; range: 0–60). The PCDAI values of the male and female participants (*p* = 0.845) were comparable but the PCDAI values correlated positively and significantly with age (r = 0.393; *p* = 0.004). The median SES-CD score was 3 (IQR 1.5–7.5); SES-CD scores correlated with neither the sex (*p* = 0.116) nor the age of the participants (r = 0.070; *p* = 0.621). Expectedly, it did correlate positively with PCDAI, although the (nonparametric) correlation was not as distinct as one might expect (r = 0.570; *p* < 0.001).

To examine the usefulness of both MRE and SBCE using objective markers of the disease, we correlated the findings with both SES-CD and PCDAI scores. Table 1 shows descriptive values for SBCE and MRE.

Table 2 shows the correlation coefficients (r) and *p*-values of the Spearman correlation between the SBCE and MRE parameters and SES-CD score. The proximal and total CECDAI values correlated positively and statistically significantly with SES-CD, while correlations for the remaining two parameters were not statistically significant.

The correlations of the same parameters with PCDAI (Table 3) were statistically significant in all four cases.

Expectedly, the correlation between the SBCE parameters and CDMI was also highly statistically significant and most pronounced for the summary score (Table 4).

Considering that both the summary CECDAI and CDMI values correlated with age (but not sex) of the participants, we constructed two simple multiple regression stepwise models to compare SBCE and MRE to PCDAI, controlling for the age of the participants.

When the summary CECDAI value was considered a predictor, the stepwise method excluded age from the final model and an R2 value of 0.337 was reached, with the summary CECDAI score as the only independent variable. In a similar model but with CDMI as a predictor, the R2 value was only marginally lower, with a value of 0.332, with age also excluded from this model. Both models were statistically significant at *p* < 0.001 and in conclusion, both summary CECDAI and CDMI values were remarkably similar in their role as predictors of the PCDAI as the objective parameter of the disease.

## 4. Discussion

The management of Crohn’s disease in pediatric patients poses significant therapeutic challenges due to the disease’s potential lifelong duration. Early and precise diagnosis is crucial to initiate timely treatment and improve long-term outcomes. Diagnostic methods vary based on the symptoms and disease extent, necessitating a comprehensive approach. This study aimed to evaluate the effectiveness of Small Bowel Capsule Endoscopy (SBCE) and Magnetic Resonance Imaging (MRI) in diagnosing and monitoring Crohn’s disease in pediatric patients using the Pediatric Crohn’s Disease Activity Index (PCDAI) and the Simple Endoscopic Score for Crohn’s Disease (SES-CD) as objective measures of disease activity.

Our study demonstrated that both SBCE and MRI correlated positively and statistically significantly with PCDAI scores, indicating their utility in assessing disease activity. While the correlations with SES-CD were less pronounced, both imaging modalities showed similar efficacies when assessing disease activity via PCDAI. This suggests that SBCE and MRI are comparable tools for diagnosing and monitoring Crohn’s disease in pediatric patients, with SBCE offering a non-invasive alternative that is well-tolerated by patients. The non-invasive nature of SBCE is particularly beneficial in pediatric patients, as it reduces the need for the sedation and radiation exposure associated with other imaging techniques. A study by Cohen et al. demonstrated that SBCE significantly impacts medical decision-making and treatment strategies in pediatric Crohn’s disease, leading to changes in management in a substantial proportion of cases [24].

In the context of the existing literature, our findings support the growing evidence that SBCE is a valuable diagnostic tool for Crohn’s disease. SBCE provides detailed mucosal visualization without radiation or contrast, addressing some of the limitations of traditional imaging methods like barium studies and magnetic enteroclysis. Studies have shown that SBCE is superior to MRE in detecting small bowel lesions, particularly superficial and proximal lesions, with a higher diagnostic yield in detecting jejunum and ileum involvement. For instance, Jensen et al. highlighted the superiority of SBCE in detecting small bowel lesions compared to other imaging modalities, emphasizing its role in diagnosing Crohn’s disease in areas difficult to reach with traditional endoscopy.

Additionally, SBCE has been found to significantly impact medical decision-making and treatment strategies in pediatric Crohn’s disease, leading to changes in management in a substantial proportion of cases. A study by Cohen et al. demonstrated that SBCE findings influenced treatment decisions in over 70% of pediatric patients, underscoring its clinical utility in guiding therapy [24]. This impact on treatment decisions is crucial, as it can lead to earlier initiation of the appropriate therapy, potentially improving the disease outcomes.

One unexpected finding was the relatively lower correlation of both SBCE and MRI with SES-CD compared to PCDAI. This discrepancy may be attributed to the different aspects of disease activity that each index measures. PCDAI assesses clinical symptoms and growth parameters, while SES-CD focuses on endoscopic severity. This highlights the importance of using a combination of clinical and endoscopic assessments to fully understand disease activity. For instance, magnetic resonance enterography (MRE) scores, such as MaRIA and Clermont, have been shown to correlate with endoscopic indices and can be used to monitor disease changes in children with Crohn’s disease undergoing induction treatment. A study by Radhakrishnan et al. demonstrated that MRE-based scores correlate with clinical activity in pediatric Crohn’s disease, offering a valuable tool for disease management [25].

Our study had several limitations. The small sample size and the lack of long-term follow-up data limit the generalizability of our findings. Additionally, the high cost of SBCE and the need for specialized equipment for MRI are significant barriers to widespread adoption. Future studies should aim to address these limitations by exploring cost-effective strategies and validating the long-term benefits of these diagnostic methods. Furthermore, integrating SBCE findings with clinical, endoscopic, and histological data is crucial to avoid misinterpretation of drug-related lesions, emphasizing the need for comprehensive diagnostic approaches. Fecal calprotectin, a biomarker of intestinal inflammation, has been proposed as a tool to predict SBCE findings and optimize the use of endoscopic procedures in pediatric Crohn’s disease. A study by Henderson et al. found that fecal calprotectin levels correlated with SBCE findings, suggesting its potential as a non-invasive screening tool to identify patients who may benefit from further endoscopic evaluation [26].

The implications of our study are significant for clinical practice. SBCE offers a viable alternative to MRI for diagnosing and monitoring Crohn’s disease in pediatric patients, enabling earlier initiation of the proper therapy and potentially improving disease outcomes. As SBCE technology evolves, education and experience among clinicians will be essential to enhance its sensitivity and specificity. Recent advancements in image-enhanced capsule endoscopy have shown promise in improving the detection of small intestinal lesions, potentially enhancing the diagnostic yield of SBCE. Recommendations for future research include investigating the cost-effectiveness of SBCE compared to MRI and exploring the integration of emerging technologies to improve diagnostic accuracy.

## 5. Conclusions

In conclusion, our study supports the use of SBCE as a non-invasive and effective diagnostic tool for Crohn’s disease in pediatric patients, which was comparable to MRI in assessing disease activity. By addressing the limitations of the current diagnostic methods and integrating SBCE into clinical practice, we can improve the management of Crohn’s disease and enhance patient outcomes. Future research should focus on overcoming the current barriers to the widespread adoption of SBCE and exploring its long-term benefits in managing this chronic condition.

## Figures and Tables

**Figure 1 biomedicines-13-01157-f001:**
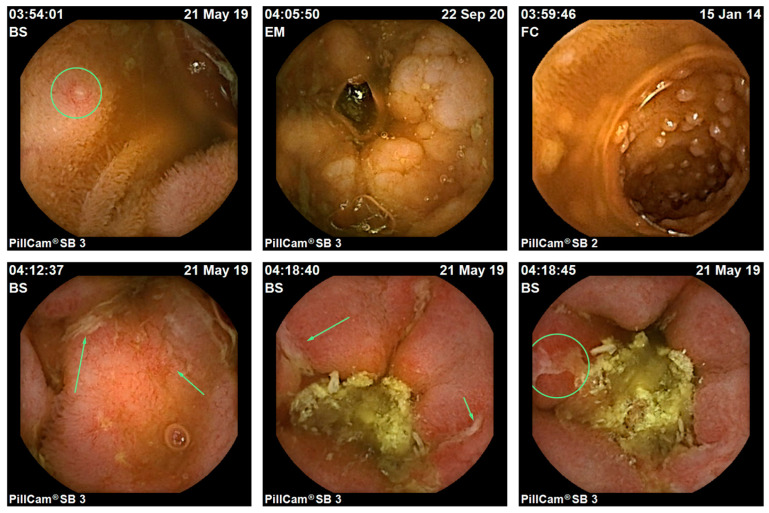
Clearly displayed lesions (arrows and circles) delineated for score calculation.

**Figure 2 biomedicines-13-01157-f002:**
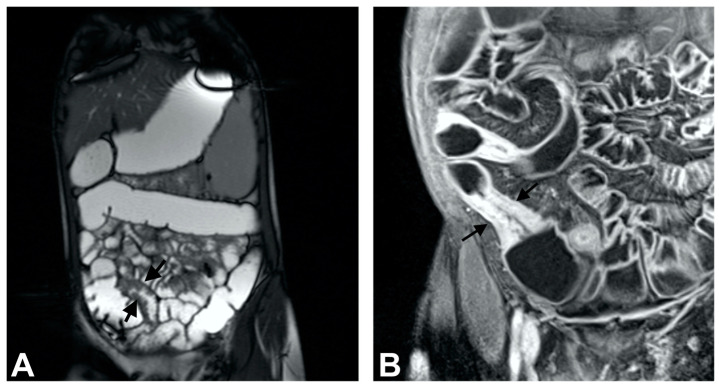
T2-weighted coronal scan reveals nodular mucosa (“cobblestone”) of the terminal ileum as a sign of mild inflammation (**A**; arrows); Postcontrast T1-weighted axial scan with fat suppression demonstrates skip lesions of the small bowel with intensive enhancement, wall thickening, and stratification (**B**; arrows).

**Table 1 biomedicines-13-01157-t001:** Descriptive parameters for SBCE and MRE.

Parameter	N *	M	Med	Min	Max	LQ	UQ	SD
CECDAI prox	52	2.10	0.00	0.00	13.00	0.00	4.00	3.46
CECDAI dist	52	4.27	4.00	0.00	13.00	2.00	6.50	3.59
CECDAI sum	52	6.37	4.00	0.00	26.00	2.00	8.50	6.20
CDMI	52	1.33	0.00	0.00	7.00	0.00	2.00	2.00

* N = number of subjects; M = mean value of the parameter; Med = median; Min/Max = minimum and maximum observed values; LQ/UQ = lower and upper quartiles; SD = standard deviation.

**Table 2 biomedicines-13-01157-t002:** Correlations between SBCE and MRE parameters and SES-CD.

Parameter	r	*p*
CECDAI prox	0.337	0.015
CECDAI dist	0.191	0.176
CECDAI sum	0.336	0.019
CDMI	0.205	0.145

**Table 3 biomedicines-13-01157-t003:** Correlations between SBCE and MRE parameters and PCDAI.

Parameter	r	*p*
CECDAI prox	0.504	<0.001
CECDAI dist	0.355	0.010
CECDAI sum	0.517	<0.001
CDMI	0.525	<0.001

**Table 4 biomedicines-13-01157-t004:** Correlations between SBCE parameters and CDMI.

Parameter	r	*p*
CECDAI prox	0.511	<0.001
CECDAI dist	0.529	<0.001
CECDAI sum	0.605	<0.001

## Data Availability

The original contributions presented in this study are included in the article. Further inquiries can be directed to the corresponding author.

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
