# Peer review of "Accuracy of Imaging Scoring Indexes in Pediatric Crohn’s Disease Patients"

_biomedicines, 2025, doi:10.3390/biomedicines13051157_

Round 1
Reviewer 1 Report
Comments and Suggestions for Authors
Diagnosing and monitoring the activity of Crohn’s disease is challenging especially in paediatric patients. This manuscript employed and compared two techniques including small bowel capsule endoscopy (SBCE) and magnetic resonance enterography (MRE) in assessing the paediatic CD using methodologies Capsule Endoscopy Crohn’s Disease Activity Index (CECDAI) and the Crohn’s Disease MRI Index (CDMI), respectively. This study found that SBCE seems to be a more reliable, non-invasive tool than MRE with higher resolution for mucosal evaluation. The study may add important updates to disease diagnosis for current Crohn’s disease in paediatrics. However, some concerns should be addressed before it can be accepted. Below are my comments.
Major comments
- Because Crohn’s disease affects gastrointestinal tract either partially or as a whole, how do authors make sure that two techniques SBCE and MRE accurately locate the lesion at same or similar positions when comparing them? For comparison, if the lesion positions are not considered, what practical values can two techniques provide? Will the lesion positions significantly affect the evaluation results for two described techniques?
- In the methods part, there is no information about the body weight of those participants. I am concerning if overweight participants will cause significant difference in diagnosis results of Crohn’s disease. Do authors have any information about this aspect?
- I understand that this study was designed for Crohn’s disease in pediatrics. How do authors think if adult patients can be used in related studies?
Minor comments
- Line 19 about the “Objective”, it can be revised to make it clearer and more understandable.
- Authors introduced PCDAI and SES-CD, but actually I cannot find details about them. I suggest that at least they should be briefly described what they are in the introduction part.
- Line 51, what is “colitis2”? Correction needed?
Author Response
Comment 1:
Because Crohn’s disease affects gastrointestinal tract either partially or as a whole, how do authors make sure that two techniques SBCE and MRE accurately locate the lesion at same or similar positions when comparing them? For comparison, if the lesion positions are not considered, what practical values can two techniques provide? Will the lesion positions significantly affect the evaluation results for two described techniques?
Response 1:
We thank the reviewer for raising this important consideration. To ensure accurate lesion localization between Small Bowel Capsule Endoscopy (SBCE) and Magnetic Resonance Enterography (MRE), lesions are usually categorized based on established anatomical landmarks (e.g., proximal, mid, and distal small bowel segments). This standardized classification allowed meaningful comparisons of lesion detection and localization between techniques.
Even without exact lesion positional alignment, SBCE and MRE provide complementary diagnostic information. SBCE excels in visualizing mucosal detail and identifying superficial lesions or subtle mucosal changes, whereas MRE provides critical insight into transmural inflammation, bowel-wall thickening, and extraluminal complications (e.g., abscesses, fistulae).
While precise lesion positions can enhance correlation analyses, minor discrepancies in lesion localization generally do not significantly alter the comparative value of these techniques, especially when evaluating overall disease extent, activity, and complications. Therefore, both modalities retain practical clinical value despite potential variations in lesion localization.
Additionally, this is a question we ourselves encountered during our years of work on this topic. While it certainly makes sense to question the comparability of the spatial distributions of lesions detected by different techniques, our experience indicates that, statistically speaking, the probability of involvement of a specific bowel segment detected by various methods is actually comparable.
Comment 2:
In the methods part, there is no information about the body weight of those participants. I am concerning if overweight participants will cause significant difference in diagnosis results of Crohn’s disease. Do authors have any information about this aspect?
Response 2:
While specific body weight or BMI data were not systematically recorded for this study, our patient group represents a typical clinical population, randomly selected, without any notable or visible extremes in body mass index (BMI). Based on clinical experience, moderate variations in BMI are unlikely to significantly influence the diagnostic performance of SBCE or MRE in Crohn’s disease. Nevertheless, we acknowledge this limitation explicitly in the revised manuscript and recommend addressing it with more detailed anthropometric data in future studies to better explore the potential impact of body weight.
Comment 3:
I understand that this study was designed for Crohn’s disease in pediatrics. How do authors think if adult patients can be used in related studies?
Response 3:
Although our study was specifically designed for pediatric patients with Crohn’s disease, we believe the diagnostic methodologies (SBCE and MRE) could be effectively translated to adult populations. Indeed, the fundamental imaging principles remain consistent across age groups. However, given the differences in disease progression, anatomical considerations, and potential comorbidities between pediatric and adult patients, validation in adult populations would be beneficial. We have addressed this point briefly in the revised discussion to highlight potential future research directions.
Minor comment 1:
Line 19 about the “Objective”, it can be revised to make it clearer and more understandable.
Response:
Addressed in the manuscript.
Minor comment 2:
Authors introduced PCDAI and SES-CD, but actually I cannot find details about them. I suggest that at least they should be briefly described what they are in the introduction part.
Response:
We added additional text to the Subjects and Methods section.
Minor comment 3:
Line 51, what is “colitis2”? Correction needed?
Response:
It was an error, changed now.
Reviewer 2 Report
Comments and Suggestions for Authors
Dear Editors and authors,
1-Through my review of some previous studies, I found that the current manuscript is very similar to the manuscript mentioned in
https://www.thieme-connect.com/products/ejournals/html/10.1055/s-0038-1637087
Knowing that some of the authors are the same ones who wrote the manuscript referred to and with the same goals and almost the same work steps. What is the scientific benefit of repeating the same results?
2-The authors did not mention the sample size of the experiment, the number of patients who underwent the experiment, how many people what are their details their ages their genders? Everything is vague about them?
3-Figure 1 and Figure 2: Did the authors do their work or were they taken from the hospital?
4-In the present study, we enrolled 52 subjects, 35 boys and 17 girls. In line 171 we note the number of subjects (patients) of the experiments and their details, and this information should be mentioned in the methods of work, not the results.
5-The symbols mentioned in Table 1 must have their meanings stated below the table.
7-The discussion is very weak. No previous manuscript or study was mentioned and the results were not compared with previous studies. In general, I referred to a study similar to the current study by the same author and the results of the two studies were not compared.
8-There are no conclusions in the manuscript, and this is what was mentioned in the Conclusions section (This section is not mandatory but can be added to the manuscript if the discussion is unusually long or complex.)
Author Response
Comment 1:
Through my review of some previous studies, I found that the current manuscript is very similar to the manuscript mentioned in
https://www.thieme-connect.com/products/ejournals/html/10.1055/s-0038-1637087
Knowing that some of the authors are the same ones who wrote the manuscript referred to and with the same goals and almost the same work steps. What is the scientific benefit of repeating the same results?
Response 1:
Thank you for your comment. Please note that the work you're referring to was an oral presentation at a professional meeting, which, in terms of scope and detail, is significantly smaller compared to this study.
Comment 2:
The authors did not mention the sample size of the experiment, the number of patients who underwent the experiment, how many people what are their details their ages their genders? Everything is vague about them?
Response 2:
The data were initially in the Results section, we also mention it in the Subjects and Methods now.
Comment 3:
Figure 1 and Figure 2: Did the authors do their work or were they taken from the hospital?
Response 3:
Authors did their work within their wards at the Clinical University hospital.
Comment 4:
In the present study, we enrolled 52 subjects, 35 boys and 17 girls. In line 171 we note the number of subjects (patients) of the experiments and their details, and this information should be mentioned in the methods of work, not the results.
Response 4:
The data were initially in the Results section, we also mention it in the Subjects and Methods now.
Comment 5:
The symbols mentioned in Table 1 must have their meanings stated below the table.
Response 5:
Added the description of symbols.
Comment 6:
The discussion is very weak. No previous manuscript or study was mentioned and the results were not compared with previous studies. In general, I referred to a study similar to the current study by the same author and the results of the two studies were not compared.
Response 6:
We rewrote the discussion, added a few references as well as the Conclusion section, which was your next comment.
Reviewer 3 Report
Comments and Suggestions for Authors
Title: Accuracy of imaging scoring indexes in paediatric Crohn's disease patients
Journal: Biomedicines (ISSN 2227-9059)
Manscripit ID: biomedicines-3476420
Recommendations: Minor revisions
Comments:
Although manuscript contain new and significant information but there are some points which should be addressed
The study mentions a sample size of 52, but beyond the power analysis, there is little explanation for why this number was chosen. Was this sample size truly sufficient to detect meaningful clinical differences? Providing a stronger justification would help clarify the study’s robustness.
Additionally, the inclusion and exclusion criteria for participants could be more detailed. Clearer criteria would help readers understand the study population and any potential selection biases.
The study states that ethical approval was obtained in May 2011, yet data collection continued until September 2020. Was there any reason an updated ethical approval was not obtained closer to the actual study period? This could be important for ensuring compliance with evolving ethical standards.
Some of the references, particularly key epidemiological data, seem outdated. Incorporating more recent sources from 2020–2024 would strengthen the study’s relevance.
The methodology states that this was a "prospective double-blinded comparison study," but it’s unclear what exactly was blinded? Clarifying this would improve transparency.
The section on SBCE image interpretation and standardization needs more details.
It is also unclear whether the primary outcome of the study was the correlation with PCDAI or lesion detection. A clear distinction here would make the interpretation of the results much stronger.
The study does not include a control group, such as healthy subjects or patients with other gastrointestinal disorders, which could provide a stronger comparative basis for the findings.
Potential confounding variables—like disease duration, prior treatments, and diet—are not addressed, yet they could significantly impact imaging results. Including a discussion on how these factors were accounted for would be valuable.
While SBCE is described as "non-invasive," the risk of capsule retention is a known limitation that isn't adequately discussed. A brief mention of this, along with any mitigation strategies, would add balance to the findings.
For MRE, the study states that "two wrapped-around flexible surface coils" were used, but it is not clear whether this setup was the same for all patients. If there was variability, it could introduce inconsistencies in imaging results, so providing clarification would be helpful.
The results are presented as a whole without stratifying by disease severity, sex, or age group. Breaking down the findings by these factors could provide more nuanced insights.
The statistical analysis section could use more details.
Finally, while the discussion provides useful insights, it would be helpful to include clearer guidance on how these findings should influence clinical practice or patient management. This would enhance the study’s practical impact.
Author Response
Dear Sir/Madam, please find our responses in red, below. Also, please note that many of your comments have been raised (and responded to) by previous reviewers. If the reviewer thinks it is necessary to add further details/clarifications to the manuscript, we will be happy to do so.
-----
The study mentions a sample size of 52, but beyond the power analysis, there is little explanation for why this number was chosen. Was this sample size truly sufficient to detect meaningful clinical differences? Providing a stronger justification would help clarify the study’s robustness.
This was a convenience sample at our clinic. We recruited subjects basically randomly, as they came to the ward.
Additionally, the inclusion and exclusion criteria for participants could be more detailed. Clearer criteria would help readers understand the study population and any potential selection biases.
Inclusion and exclusion criteria are listed in the informed consent document, which we have uploaded separately. If the document will not be available online (such as supplemental information or similar), we can list exclusion and inclusion criteria. However, please also see the previous comment, i.e. our intention was to include the convenience sample of subjects.
The study states that ethical approval was obtained in May 2011, yet data collection continued until September 2020. Was there any reason an updated ethical approval was not obtained closer to the actual study period? This could be important for ensuring compliance with evolving ethical standards.
Since this study involved children attending our clinic as part of the routine detailed evaluation for an existing diagnosis, ethical standards remained unchanged. In other words, had this been, for example, a prospective clinical trial involving the administration of different medications, it would certainly have been necessary to re-evaluate every component of the ethics committee's decisions.
Some of the references, particularly key epidemiological data, seem outdated. Incorporating more recent sources from 2020–2024 would strengthen the study’s relevance.
We added 5 refs in the Discussion and Subjects and Methods sections.
The methodology states that this was a "prospective double-blinded comparison study," but it’s unclear what exactly was blinded? Clarifying this would improve transparency.
At lines 122-124 of the original manuscript we mention that "Data from the recorder was downloaded on site within the department of gastroenterology and analysed by one clinical SBCE expert blinded to patient clinical information (Figure 1). "
Let us know if you think this should be changed.
The section on SBCE image interpretation and standardization needs more details.
We added an additional description of the methods in the Materials and Methods section and some more detail in the Discussion.
It is also unclear whether the primary outcome of the study was the correlation with PCDAI or lesion detection. A clear distinction here would make the interpretation of the results much stronger.
We understand your point. However, due to a relatively small number of subjects at our Clinic as well as convenient sample, we felt that the primary objective should be left as described in the abstract.
The study does not include a control group, such as healthy subjects or patients with other gastrointestinal disorders, which could provide a stronger comparative basis for the findings.
We had no option of including a control group due to ethical and financial circumstances at the Clinic.
Potential confounding variables—like disease duration, prior treatments, and diet—are not addressed, yet they could significantly impact imaging results. Including a discussion on how these factors were accounted for would be valuable.
Since the patient selection was essentially random, we included only the basic demographic data (sex and age) about our subjects.
While SBCE is described as "non-invasive," the risk of capsule retention is a known limitation that isn't adequately discussed. A brief mention of this, along with any mitigation strategies, would add balance to the findings.
Thank you, included in the text!
For MRE, the study states that "two wrapped-around flexible surface coils" were used, but it is not clear whether this setup was the same for all patients. If there was variability, it could introduce inconsistencies in imaging results, so providing clarification would be helpful.
The setup was the same.
The results are presented as a whole without stratifying by disease severity, sex, or age group. Breaking down the findings by these factors could provide more nuanced insights.
We felt that the sample size was a bit too small, however, if you think such stratification could be useful, we can do additional (nonparametric) analyses to streghten the results.
The statistical analysis section could use more details.
Please see the comment above.
Finally, while the discussion provides useful insights, it would be helpful to include clearer guidance on how these findings should influence clinical practice or patient management. This would enhance the study’s practical impact.
We introduced some additional changes to the Discussion section.
Round 2
Reviewer 1 Report
Comments and Suggestions for Authors
The manuscript is improved to the level suitable for acceptance of publication.
Author Response
Thank you for your kind suggestions!
Reviewer 2 Report
Comments and Suggestions for Authors
Dear Editors and author,
In the first review, the main question was:
Through my review of some previous studies, I found that the current manuscript is very similar to the manuscript mentioned in
https://www.thieme-connect.com/products/ejournals/html/10.1055/s-0038-1637087
Knowing that some of the authors are the same ones who wrote the manuscript referred to and with the same goals and almost the same work steps. What is the scientific benefit of repeating the same results?
Response 1:
Thank you for your comment. Please note that the work you're referring to was an oral presentation at a professional meeting, which, in terms of scope and detail, is significantly smaller compared to this study.
The research concept was previously published in 2018, a study published in the same location and hospital. The sample size was 33 patients, including 14 girls.
How do the authors refer to this as a preliminary presentation only?
The three authors of the aforementioned study are present in this study.
Author Response
We thank the reviewer for the comment. Indeed, some of the results presented here were partially published earlier, in 2018. However, this involved only a brief oral presentation accompanied by a poster lasting several minutes. Here, we provide a comprehensive analysis of the same data, significantly expanded in terms of both volume and analytical depth.
We initially planned to publish the full manuscript presented here in late 2019 and early 2020. Unfortunately, the COVID-19 pandemic, the earthquake in Zagreb, and extensive teaching commitments of the authors delayed publication until now.
Although considerable time has passed, we believe that the data remain relevant and that publishing these results in this significantly expanded form remains meaningful.
Round 3
Reviewer 2 Report
Comments and Suggestions for Authors
Dear Editors,
The authors stated that this study is part of a previous study conducted in the same place. I have no objection to publishing the study, but I must ensure that nothing has been published before. It is the journal's policy that the study should not have been previously published. I leave the final decision to the journal editor.